# Description, reliability and utility of a ground-reaction-force triggered protocol for precise delivery of unilateral trip-like perturbations during gait

**Hui-Ting Shih**[1,2], **Robert Gregor**[3], **Szu-Ping Lee**[1]*

**1** Department of Physical Therapy, University of Nevada Las Vegas, Las Vegas, Nevada, United States of America, **2** Baylor Scott & White Research Institute, Dallas, Texas, United States of America, **3** School of Integrated Health Sciences, University of Nevada Las Vegas, Las Vegas, Nevada, United States of America

* szu-ping.lee@unlv.edu

**Data Availability Statement:** We provided all information to replicate the protocol as described in this paper and the supplemental information. We don't believe it is useful to include our proof-of-

## Abstract

Tripping is a common cause of falls and a focus of many biomechanical investigations. Concerns regarding the precision of delivery of simulated-fall protocols reside in the current biomechanical methodology literature. This study aimed to develop a treadmill-based protocol that generated unanticipated trip-like perturbations during walking with high timing precision. The protocol utilized a side-by-side split-belt instrumented treadmill. Programmed treadmill belt acceleration profiles (two levels of perturbation magnitude) were triggered unilaterally at the instant the tripped leg bore 20% of the body weight. Test-retest reliability of fall responses was examined in 10 participants. Utility was examined as to whether the protocol could differentiate the fall recovery responses and likelihood of falls, estimated using peak trunk flexion angle after perturbation, between young and middle-aged adults (n = 10 per group). Results showed that the perturbations could be precisely and consistently delivered during early stance phases (10–45 milliseconds after initial contact). The protocol elicited excellent reliability of responses in both perturbation magnitudes (ICC = 0.944 and 0.911). Middle-aged adults exhibited significantly greater peak trunk flexion than young adults (p = 0.035), indicating that the current protocol can be utilized in differentiating individuals with different levels of fall risks. The main limitation of the protocol is that perturbations are delivered in stance rather swing phase. This protocol addressed some issues discussed in previous "simulated fall" protocols and may be useful for future fall research and subsequent clinical interventions.

## Introduction

Falls threaten adults over 50 years of age as its prevalence has been surveyed as high as 62.1% [1]. Falls frequently happen when the normal walking cycle is disrupted, such as during a tripping event [2–4]. A previous study found that residents in long-term care facilities fell mostly due to incorrect weight-shifting (41%) and tripping (21%) [5].

concept utility data in this publication. If the readers are interested in our utility data, requests can be send to the corresponding author.

**Funding:** The work is supported by the University of Nevada, Las Vegas Physical Therapy Department (Student Opportunity Research Grant), National Institutes of Health (1K01HD091449), and International Society of Biomechanics (Matching Dissertation Award). None of the funders had any role in study design, data collection and analysis, decision to publish, or preparation of this manuscript.

**Competing interests:** The authors declare that there are no conflicts of interest.

The current consensus defines a trip as an abrupt obstruction of a foot followed by a loss of dynamic postural control during gait [6–10]. In most real-life scenarios, only one foot is obstructed. And it is this sudden stop of the progression of a leg, coupled with the unexpected forward acceleration of the body center of mass relative to the base of support that causes a trip and fall [11, 12]. Researchers have focused on trunk kinematics and kinetics in relation to leg motions and dynamic base of support to study mechanisms underlying trip-related falls and fall recovery responses [13, 14]. Pavol et al. and Owing et al. examined 39 kinematic variables during tripping in healthy older adults [13, 15]. They found that peak trunk flexion angle was not only related to but also a significant indicator of the likelihood of falls. As mounting evidence accumulated, Grabiner and Kaufman reviewed fall-related studies and identified peak trunk flexion angle as an appropriate risk biomarker for falls [16]. A recent study further showed that trunk control is a critical and task-specific parameter in avoiding falls after perturbation [17].

Currently in the literature, one of the most commonly used methodologies to simulate tripping perturbations is the treadmill deceleration/acceleration method [8, 18]. It requires no additional foot-blocking objects, and the walking surface-based perturbations can be delivered unexpectedly to minimize anticipatory reactions [11]. Moreover, the magnitude of the perturbation is easily adjustable to allow observations of a range of fall recovery responses, which is typically achieved by manipulating the duration and/or magnitude of displacement of the walking surface [19, 20]. Based on the analyses of electromyography activities and recovery responses, researchers concluded that this type of protocol was able to elicit responses comparable to what occurs in daily life [15, 18].

While treadmill-based perturbation protocols are widely used in research, several limitations have been identified, particularly regarding the precise timing of perturbation delivery. An adequate tripping protocol should allow precise control of the onset timing and perturbation duration because the trip recovery strategy performed is highly dependent on these two factors [21–23]. The timing of the perturbation onset is particularly critical since it largely determines what recovery strategy (i.e., lowering, elevating, or reaching strategy) will be performed, and the corresponding reactive responses are fundamentally different, incomparable to each other [13, 22]. Hence, there is a need for an experimental protocol that can consistently and precisely reproduce the perturbations to make valid within- or between-subject comparisons. Previous studies have exploited the walker's kinematics or a combination of kinematics and ground reaction force (GRF) as triggering conditions for perturbation delivery [24–26]. However, these studies did not focus on timing control, and therefore the precision of perturbation delivery remains unknown. For example, in the protocol described by Sessom and colleagues, the tripping perturbation was delivered to the limb in the single-limb support phase but the exact timing (i.e. early, mid, or late stance) was not prescribed [8]. Room for improvement regarding the precision and reproducibility in the fall research protocols, remains.

The purpose of this study was to develop a GRF-triggered treadmill perturbation method that emphasizes precise and reproducible timing control of the tripping perturbation delivery. Specifically, we proposed a protocol that was designed to repeatedly deliver tripping perturbations during early stance phase by referencing to GRF. In this study, we aimed to quantify the timing of perturbation delivery, establish the protocol's capability to elicit reliable fall recovery responses and tested the protocol utility by examining if it can be used to differentiate trip recovery responses between populations with different levels of fall risk (i.e., young vs. middle-aged adults).

## Materials and methods

The protocol described in this peer-reviewed article is published on protocols.io, dx.doi.org/10.17504/protocols.io.ewov1o8k7lr2/v1 and is included for printing as S1 File with this article.

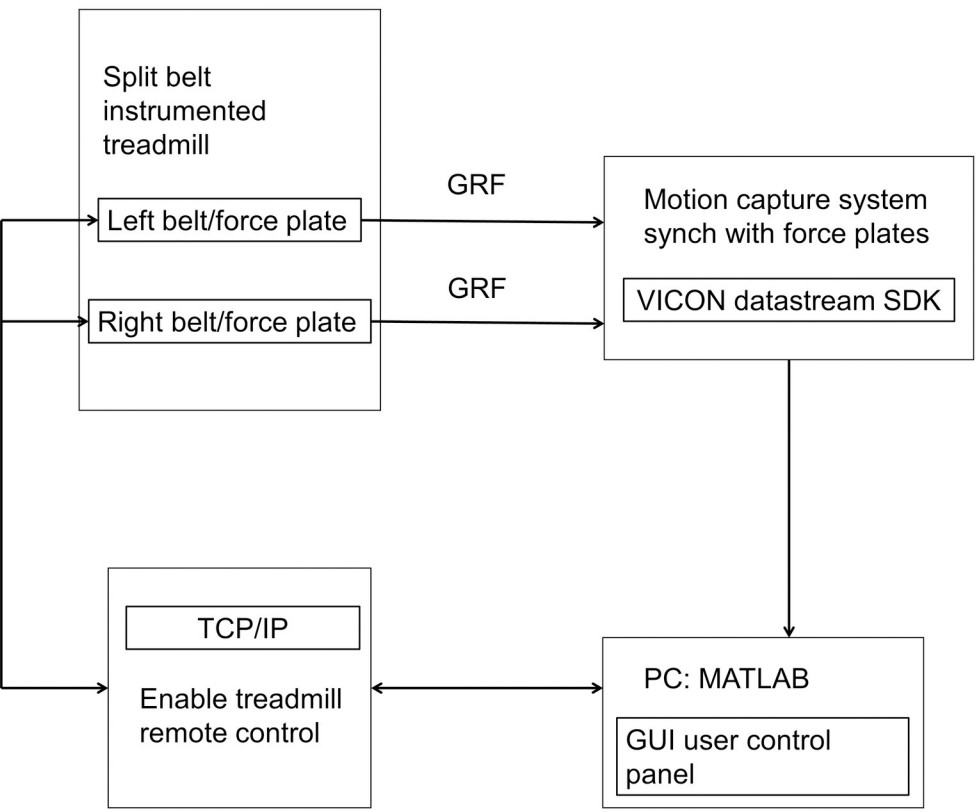

**Fig 1. System apparatus.** GRF: ground reaction force; SDK: software development kit; PC: personal computer; GUI: graphical user interface TCP/IP: transmission control protocol/internet protocol.

## System apparatus design

A Bertec side-by-side split-belt instrumented treadmill (Model ITC-11-20L-4, Bertec Corp., Columbus, OH, USA) has independent control of the movements of the two treadmill belts. Each belt is equipped with one force plate capturing GRF data from the walker's left and right foot contacts. The force data were sampled at 1000Hz, time-synchronized with the VICON (Oxford Metrics, Oxfordshire, UK) motion data and streamed by the Software Development Kit (SDK) to MATLAB (MathWorks Inc., Natick, MA, USA) on a personal computer. The MATLAB code (Supporting Information) served as the interface to communicate between Datastream SDK and the treadmill controller. Concisely, the program read the vertical GRF (vGRF) from Datastream SDK and executed the pre-programmed perturbations via the treadmill controller. The treadmill controller received remote control commands from MATLAB via the Transmission Control Protocol/Internet Protocol port, through which the program delivered the prescribed tripping perturbation by accelerating/decelerating the treadmill motors (Figs 1 and 2B).

## Perturbation design

The treadmill-based tripping perturbation protocol began with establishing participants' comfortable walking speeds (CWS). Participants were asked to self-select a walking speed during a 1-minute treadmill walking trial. The speed was finetuned by participant's request or adjusted if the participant did not walk in the middle of the treadmill. We increased the speed

**A.**

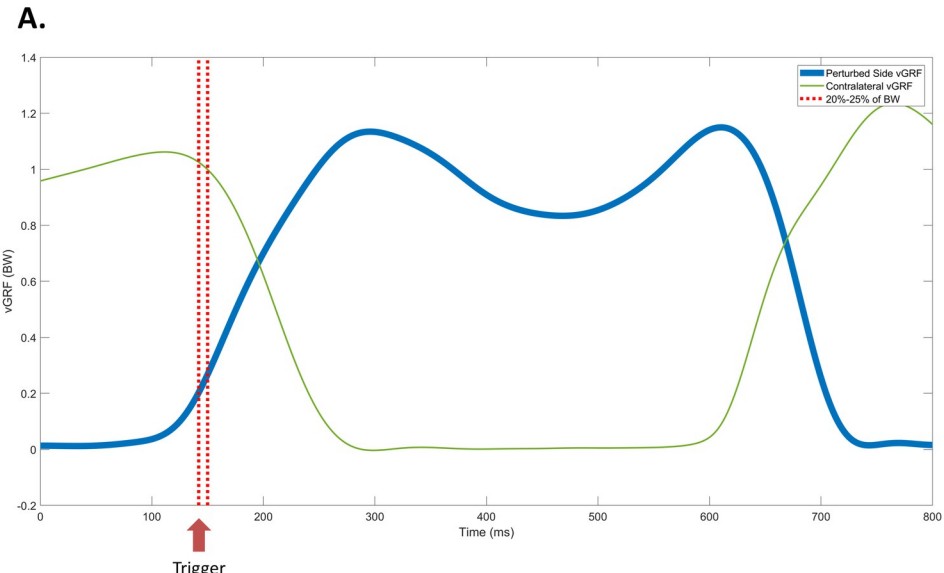

**B.**

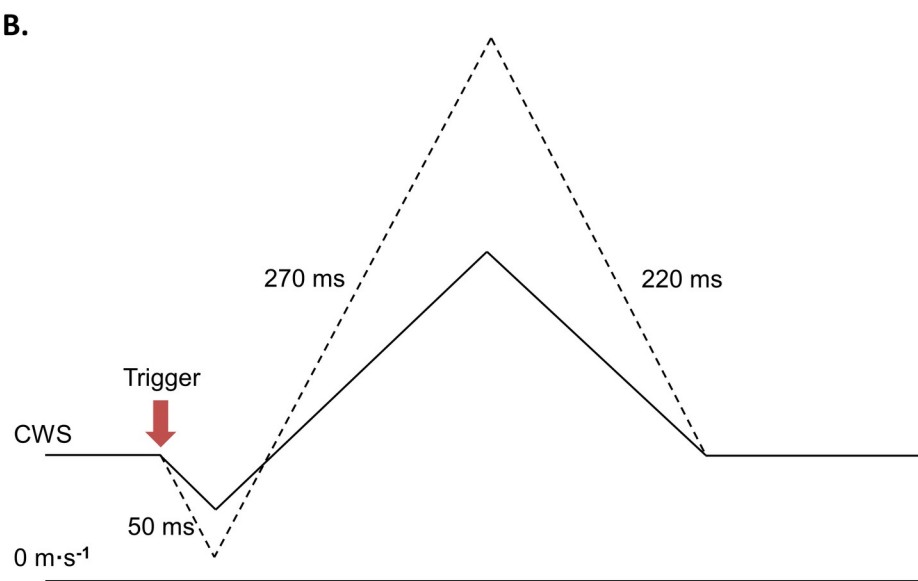

**Fig 2. Perturbation triggering criteria and treadmill velocity profile.** (A) GRF from both limbs and perturbation triggering criteria. Blue vGRF line is from the tripped left limb; red lines enclose the designated window of a triggering event. (B) Treadmill velocity profile. Solid line stands for a tripping perturbation with a small acceleration; dash line represents a tripping perturbation with a large acceleration.

if the participant stepped to the front edge of the treadmill and vice versa, and they must be able to stay with the speed for 1 minute. During a perturbation a designated treadmill belt, either left or right, decelerated for 50 milliseconds, followed by 270 milliseconds of accelera-tion, and then decelerated again for 220 milliseconds to return to the CWS (Fig 2B). This profile was designed to simulate the sudden blockage of a foot followed by the momentary forward thrust of the body center of mass. Two acceleration levels were used to simulate trip-ping perturbations of two levels of magnitude (small vs. large) [8]. The acceleration magnitude

utilized in the protocol was linearly scaled by the CWS. For a CWS at 1 m·s⁻¹, the acceleration was either ± 6 m·s⁻² (small tripping perturbation) or ± 12 m·s⁻² (large tripping perturbation). As a constant acceleration would create a more challenging perturbation for individuals with a slower walking speed, this scale allowed the protocol to accommodate individuals with various walking speeds.

The automatic triggering criteria were based on the vGRF profile with the intention to deliver the perturbation during early stance phase of the tripped limb. We targeted early stance phase to ensure that the tripped limb went through the full course of the velocity changes of the treadmill belt, according to our preliminary data [22]. The perturbation is triggered when the following conditions are jointly met: First, the vGRF of the tripped side had to be between 20–25% of the person's body weight. Second, vGRF that met the first condition had to be greater than the vGRF 10 milliseconds prior to ensure that the trigger would occur in the ascending phase of the vGRF typical of during the early stance phase (Fig 2A).

### Electro-mechanical delay

Since the control system incorporates multiple software and hardware components, there is likely an electro-mechanical or processing delay residing in the system. We recorded the timings when MATLAB delivered the perturbation as well as the onset of treadmill belt velocity change (dashed red and solid yellow lines in Fig 3, respectively) in order to quantify the delay between these 2 components. Moreover, we confirmed whether perturbation occurred in an unintended gait phase due to the delay. Two markers were placed directly on the treadmill belts to capture the actual movement of the belts. The first frame the treadmill belt velocity ran below 2 standard deviations (SD) of the values during CWS was marked as belt velocity change onset (solid yellow line in Fig 3). The difference between these two time points was

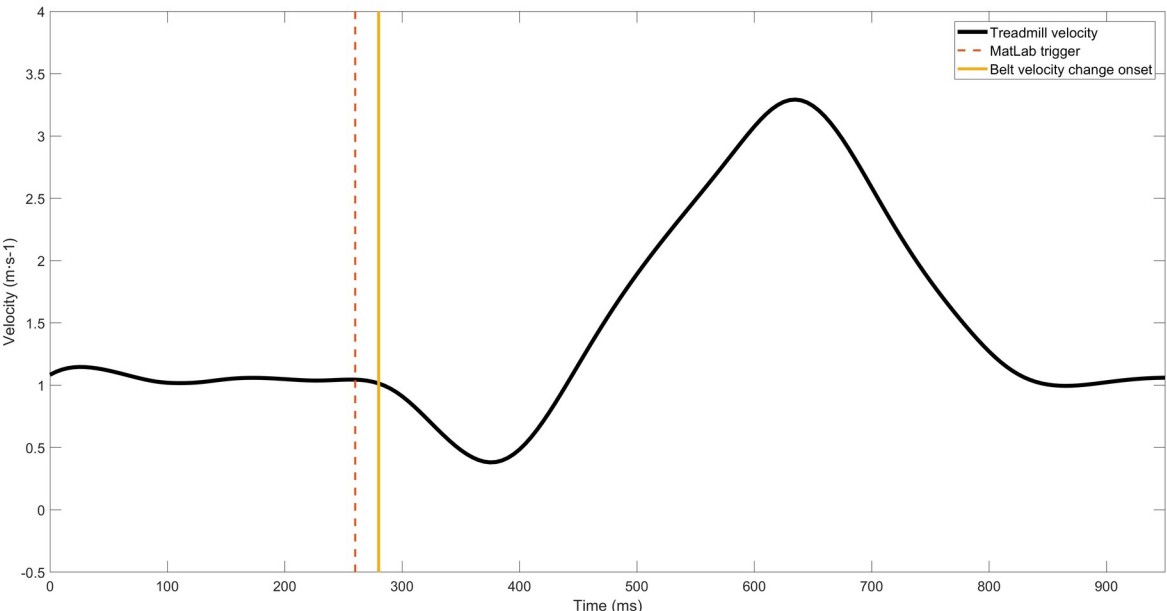

**Fig 3. Example of a trip at 12 m·s⁻² and electro-mechanical delay.** The dashed redline marks out the time that MATLB sending out the command and the solid yellow line marks the time that the treadmill belt velocity change. Latency between these two lines was the electro-mechanical delay.

defined as the delay for each trial. The foot position at the instant of perturbation delivery was also recorded to determine the perturbation timing relative to the stance phase of the perturbed limb.

## Protocol precision, reliability, and utility

We examined the following parameters: 1) timing consistency in delivering the perturbation; 2) test-retest reliability of the fall recovery responses elicited by the protocol; and 3) utility defined as whether the protocol can be used to differentiate the trip recovery trunk kinematic responses between young and middle-aged to older adults. The research protocol and procedures were approved by the Biomedical Institution Review Board of University of Nevada, Las Vegas.

To confirm the precision and consistency of perturbation delivery timing, the onset timing was calculated as a percentage of stance phase. The length of the stance phase was obtained from the normal walking period of each trial before the first perturbation was delivered. Since perturbations were delivered unilaterally, an average value of 3 strides of each leg was computed for calculations of its corresponding side of perturbation.

For reliability, we recruited participants from a university student population with no known diseases, injuries, or impairments that may influence their gait. Participants for reliability tests visited the laboratory twice, two weeks apart, and we compared peak trunk flexion angles between the two visits (Fig 4). Procedures during the two visits were identical. After providing consents for participation, participants were tested with the abovementioned protocol with the kinematics captured using a 12-camera VICON motion capture system (sampling rate = 200Hz). A marker set for lower extremities and trunk was applied by the same investigator (Fig 5A). A harness attaching to a load cell was adjusted to prevent the participants from hitting the ground without interrupting their gait (Fig 5B). Four different tripping perturbations were delivered, including: small perturbations to the left limb and to the right limb, large perturbations to the left limb and to the right limb. Perturbations were delivered in random order of timing, magnitude, and side. The durations of the walking trials varied person by person, generally less than 5 minutes. A few situations could make the duration longer. First, participants requested breaks between perturbations. Second, when a person fell and could not

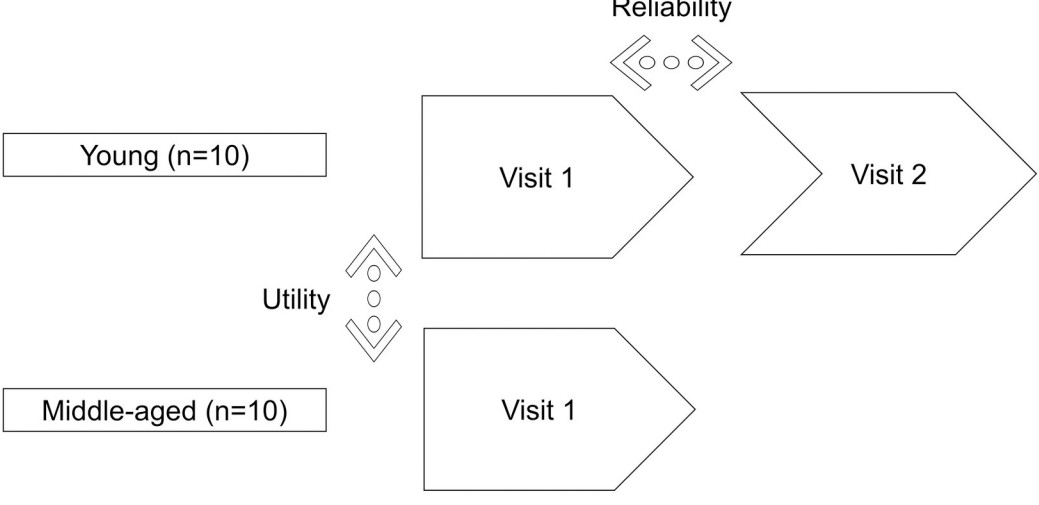

**Fig 4. Experiment procedure.**

A.

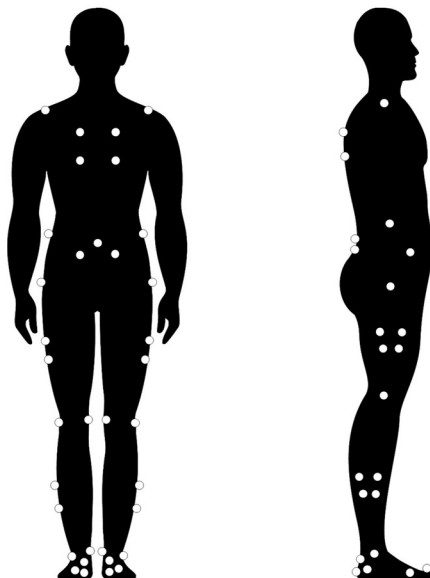

B.

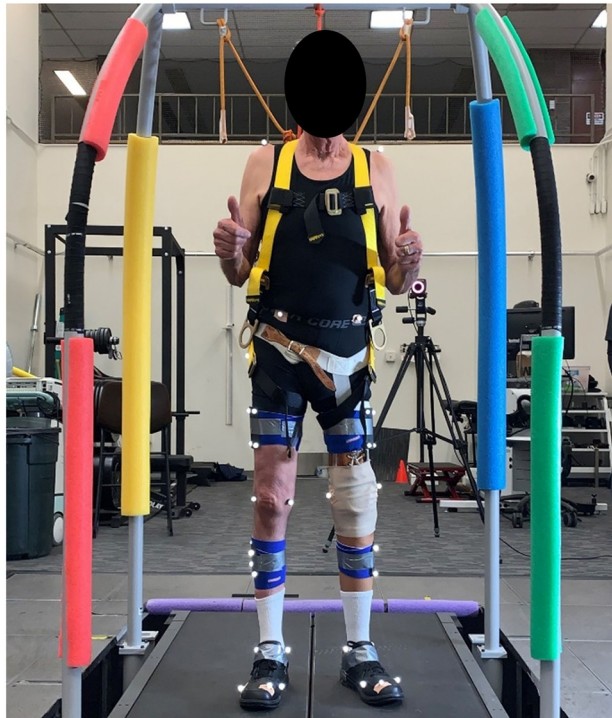

**Fig 5. Experimental settings.** (A) Passive reflective marker set. (B) Hardware settings.

return to walk, we stopped the treadmill, repositioned the participant, and completed the rest of the trials. Third, after a perturbation, we ensured by observation that they walked steadily again before delivering the next perturbation, and the time each person needed to resume a steady gait was different.

A group of middle-aged to older participants (>50 years of age) was recruited from the university and local community for the purpose of protocol utility. They had no neurological, cardiovascular, and current musculoskeletal diseases that would preclude them from walking on the treadmill. Previous reports have shown that older individuals showed attenuated responses in thigh muscle activation and greater trunk flexion displacement in a forward loss of balance scenario when compared to young cohorts [27, 28]. Given the known differences in these two age groups, we tested the utility of our protocol by examining if the reproduced trip-like scenario differentiated the two groups in terms of trunk kinematics. The middle-aged group visited the laboratory once and went through the same procedures described above. We compared their trunk kinematic responses against the values from young participants' first visit to eliminate the possibility of learning effects (Fig 5).

### Dependent variables

Peak trunk flexion angle during a tripping event has been shown to be a predictor of falls [15, 29]. Three events, perturbation onset, contralateral limb contact, and the tripped limb contact were chronologically identified in reference to foot position and vGRF magnitude. The kinematic data were lowpass-filtered at 10Hz. Positive trunk angles indicate trunk flexion. Peak trunk flexion angle, obtained after perturbation onset and before tripped limb contact, was extracted. Each participant's left and right tripping trials at the same acceleration were averaged and used for analysis. Falls were defined as when the participants had to grab the supporting struts, or when the load cell/harness supported more than 50% of their body weight. Fall trials were removed from kinematic analysis.

### Sample size estimation and statistical analysis

Electro-mechanical delay is presented with mean, SD, and range. Test-retest reliability of the peak trunk flexion angle was estimated by intraclass correlation coefficients (ICC). We considered the test-retest reliability of our primary purpose and used a hypothesis testing approach to calculate the sample size needed [30]. The hypothesis testing approach is used when the goal of the study is to show that the ICC is larger than a certain value of interest, which is commonly the goal of reliability studies in exercise, sport, and rehabilitation [31]. For our study, we determined the minimum threshold of ICC as 0.7 for we hope our tool shown to have sufficiently high reliability. The anticipated ICC was set at 0.95, power at 80%, and significant level at 0.05, and the number of repeats was 2. The estimated sample size was 10. Two-way mixed-effects model with absolute agreement definition for single measurement was selected ($ICC_{3,1}$) [32]. Two-way ANOVAs were used for examining the effects of group (young vs. middle-aged) and perturbation accelerations (small vs. large) on peak trunk flexion angle. Level of significance was set at 0.05. Statistical analyses were all conducted using SPSS version 24 (IBM Corp., Armonk, NY, USA).

### Expected results

Ten young individuals and twelve middle-aged to older individuals participated in this study. One person in the middle-aged group reported hip pain and another expressed fear of falling after the first tripping perturbation. These two individuals withdrew from the study and were not included in the analysis (Table 1). One example of trunk flexion angle during perturbation

**Table 1. Demographics.**

|  | Age (y) | Sex (M/F) | BW (kg) | CWS (m·s⁻¹) |
|---|---|---|---|---|
| Young (n = 10) | 20.90±1.66 | 3/7 | 62.79±9.63* | 0.90±0.17 |
| Middle-aged (n = 10) | 57.10±4.70 | 7/3 | 84.20±12.96* | 0.95±0.28 |

y, years; M, male; F, female; BW, body weight; CWS, comfortable walking speed.

* denotes significant difference between groups.

and recovery was shown in Fig 6. For capturing electro-mechanical delay, 24 trips were collected and analyzed. The duration of delay was, on average, 26.5±7.74 (range [5–35]) milliseconds. Perturbations were consistently delivered at 10–45 milliseconds after initial contact, which corresponds to approximately 2.5% (median, range [1.42–4.55%]) of the stance phase (Fig 7).

## Test-retest reliability

Peak trunk flexion angles during both the small and large perturbations were reliability elicited between the visits (Table 2). The protocol yielded excellent test-retest reliability in terms of reproducing trunk flexion angle during both small (ICC = 0.944, 95% CI [0.792, 0.986]) and large perturbations (ICC = 0.901, 95% CI [0.650, 0.975]).

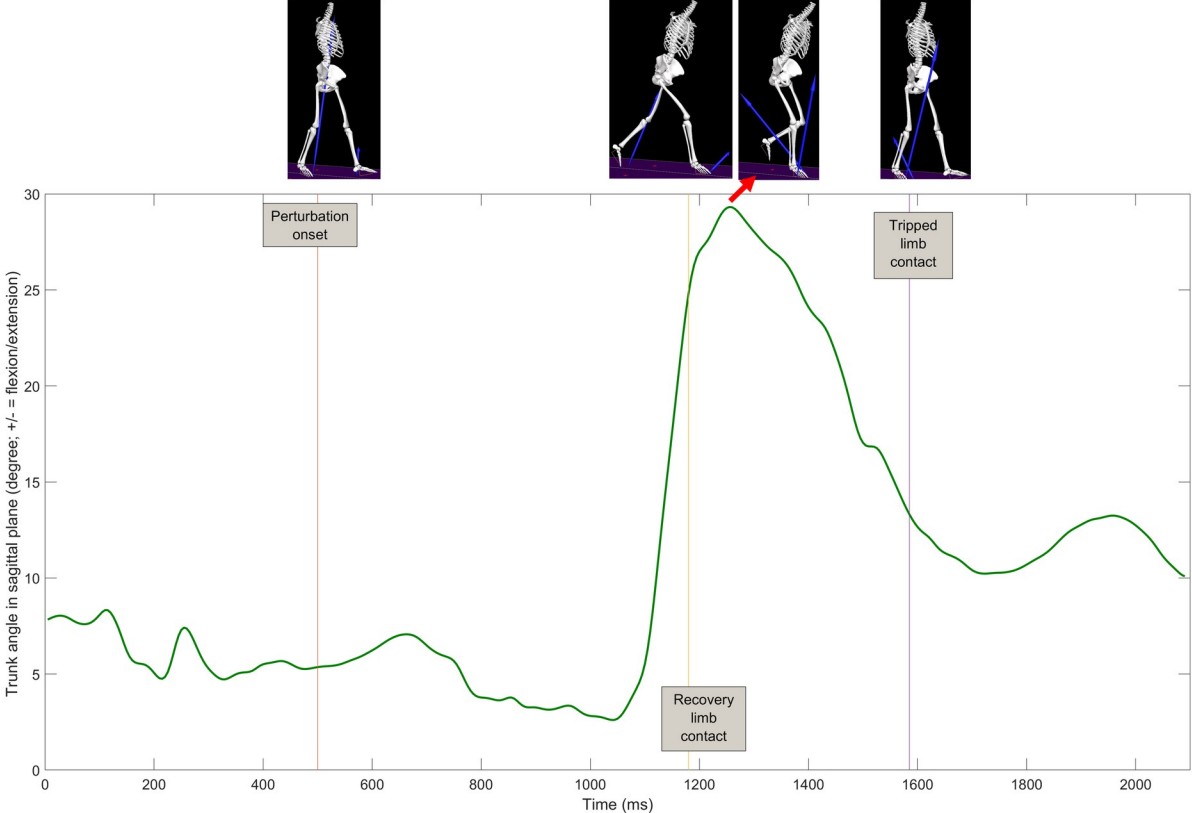

**Fig 6. Example of sagittal trunk flexion angle in response to the tripping perturbation and recovery.** Left limb was the tripped limb.

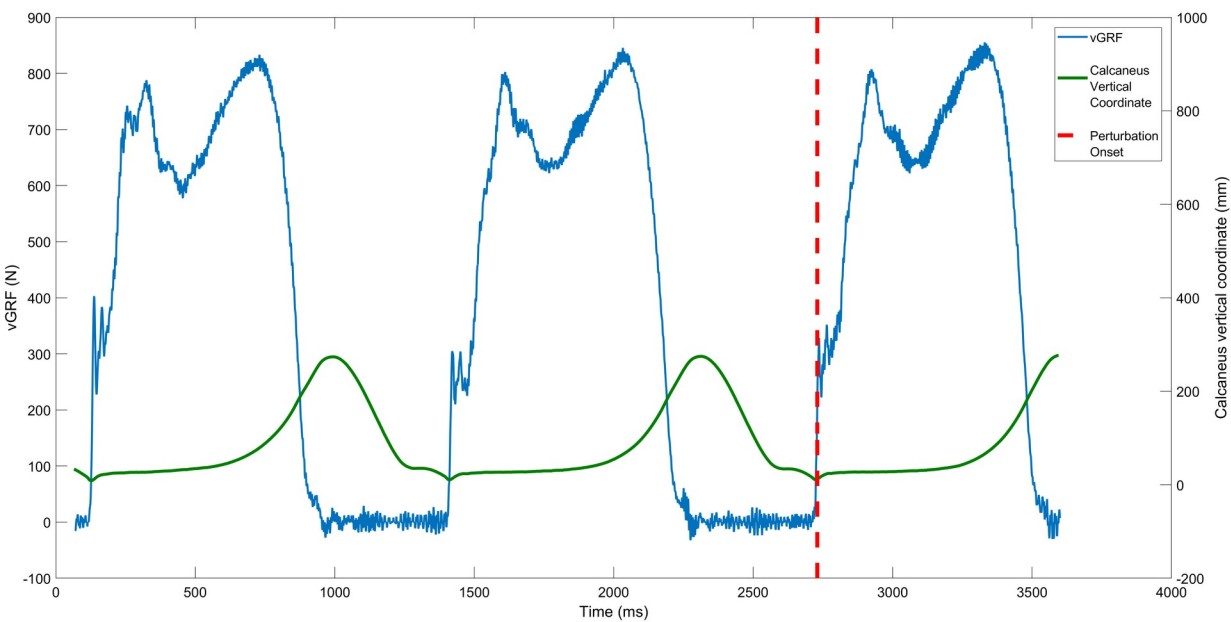

**Fig 7. Real-time raw GRF and calcaneus position of tripped limb at one perturbation.**

## Protocol utility

Of the 20 included participants, the middle-aged group was significantly heavier ($p<0.001$; Table 1). There was no significant difference in CWS between groups ($p = 0.63$). None of the participants in the young group fell in any tripping trials. The middle-aged group had 3 falls under the small perturbation condition and 6 falls under large perturbation condition, yielding percentages of fall 15% and 30% respectively (Table 2). Between the young and middle-aged groups, significant group main effect ($p = 0.035$) and significant perturbation magnitude main effect ($p<0.001$) in peak trunk flexion angle were detected. Specifically, middle-aged adults exhibited 45% greater peak trunk flexion angle when compared to young adults (Table 2). Large perturbations elicited 72% greater peak trunk flexion angles than small perturbations.

## Data discussion

We confirm the protocol's ability in delivering perturbations at precise times and eliciting reliable kinematic responses. First, we observe that all perturbations are precisely delivered before the first 5% of the stance phase as intended in a range of different CWS and stance phase durations. This high level of consistency is attributed to the GRF-based triggering criteria. Second,

**Table 2. Fall incidence and peak trunk flexion angle from young and middle-aged groups, during small and large tripping perturbations.**

| | Small acceleration | | | Large acceleration | | |
|---|---|---|---|---|---|---|
| | Fall frequency (%) | Peak trunk flexion angle (°) | | Fall frequency (%) | Peak trunk flexion angle (°) | |
| | | Visit 1 | Visit 2 | | Visit 1 | Visit 2 |
| Young (n = 10) | 0 (0%) | 13.37±6.89 | 13.55±6.72 | 0 (0%) | 23.50±10.00[†] | 23.20±8.19 |
| Middle-aged (n = 10) | 3 (15%) | 19.88±6.44* | | 6 (30%) | 33.70±10.05*[†] | |

* denotes significant group main effect between young and middle-aged participants.

[†] denotes significant perturbation magnitude main effect between large and small accelerations.

the test-retest reliability of the protocol is confirmed by participants' reproducible trunk kinematics in response to the tripping perturbation. With a two-week interval between tests, this protocol is able to elicit similar and reliable test-retest responses from the participants. A protocol that provides precise and reproducible perturbation timing will enable further investigation into the mechanisms of falls and recovery.

A treadmill-based protocol has the strength of minimizing anticipation before perturbations and eliminating movement disturbance after perturbations. In previously established simulated tripping research protocols based on obstacles, a concealed object was either placed on the ground to obstruct the foot or dropped onto the treadmill in order to produce a trip [6, 27, 33]. There has been the concern of anticipation due to that the participants might see the obstacle out of the corner of their eyes, hear the sound, perceive the vibration made by the obstacle. A treadmill-based protocol is free from the concern because no physical obstacle is needed. There were other protocols utilizing a cable attached to a person's ankle/foot to simulate tripping perturbations. Researchers tightened the cable to interfere the gait as a perturbation, but the taut cable might also interfere the natural responses after the perturbation [34]. In a treadmill-based protocol, nothing attaches to our participants' foot so that the post-perturbation responses will not be disturbed. A main limitation of this protocol is that the tripping perturbations are delivered during stance as opposed to the swing phase. This is an inherent limitation of the treadmill-based tripping protocol given that perturbation can only be delivered to the leg in stance. However, as proven before, a treadmill-based protocol can elicit comparable responses to other methods that deliver perturbations in swing. Without needing additional instruments (i.e., obstacle and cable) and worries about anticipation, our success in delivering a perturbation in a precise, reliable manner supports its future usage.

Our protocol uses GRF-based triggering criteria to precisely invoke the trip by taking advantage of the modern force plate-instrumented treadmills. GRF has been shown to remain consistent within one's gait cycle and most consistent at one's self-selected walking-speed [35, 36]. Our protocol delivers trips triggered by GRF while the participant walks at a self-selected CWS. In doing so, the tripping perturbations were induced precisely at the same phase in the gait cycle (i.e., the early stance phase) and elicited consistent trunk kinematic responses. Therefore, our data support that the novel GRF-based triggering protocol can deliver reproducible tripping perturbation results due to the precise timing control.

Precision brings in two advantages to our protocol: reproducibility and mitigating learning effect. The onset timing of a tripping perturbation affects the corresponding recovery strategy [22]. Moreover, it is challenging to compare different gross recovery strategies (i.e., elevating and lowering). The capability of repeatedly applying perturbations at a precise, consistent time window allows reproducing comparable recovery responses. Parallelly, unsuccessful trials (trials that do not elicit wanted, comparable recovery strategy) are minimized. As a result, our novel protocol with more consistent responses reduces the need of multiple trials and mitigates the confounding learning effect caused by repeated exposure.

Another feature of our protocol is unilateral foot perturbations enabled by the use of a side-by-side dual belt treadmill. Berger et al. developed the idea of simulated tripping during walking by manipulating the treadmill belt velocity [18] which overcame the abovementioned difficulty in obstacle- and cable-based protocols. However, most existing treadmill-based protocols perturbed both feet simultaneously while tripping in real-life is unilateral. To achieve unilateral perturbations, Sessoms et al. developed a protocol in which the tripping perturbation was delivered in the single-limb support phase (i.e., near mid-stance) [8]. However, we noticed two issues. First, being perturbed in the single-limb support phase indicated that the tripped limb bore a significant percentage of the body weight at the instance of tripping onset, which is a rare case in reality. Second, perturbation delivery close to mid-stance is potentially problematic

when examining individuals with disabilities since their double stance phase is longer and may not achieve true single limb support due to their slower gait velocity. The two previous studies also did not clearly describe the triggering criteria and were not specifically designed to target a certain instant of gait. Conversely, our protocol successfully delivers perturbations to a small window during gait via clear criteria of GRF.

In the current protocol, two treadmill belt acceleration profiles are utilized to simulate two levels of magnitude of tripping perturbations. We observed the magnitude main effect on peak trunk flexion angles in both age groups. In response to the larger perturbations, both groups exhibited greater peak trunk flexion angles. Our results aligned with the previous findings that more severe falls is associated with a loss of trunk control [37, 38]. The magnitude main effect we found also echoed with the results by Lee et al. who examined the slip recovery responses after different severities of perturbation [39]. Altering the duration of the tripping perturbation may be another way to manipulate the magnitude of perturbations [40]. However, manipulating the duration of the perturbation may influences the gross fall recovery strategy selection [23]. The acceleration profile can easily be manipulated in our novel protocol. Researchers and clinicians can leverage this feature to provide different levels of perturbation to participants/patients [41].

Our results suggest that the proposed protocol could be an assessment tool to differentiate the trip recovery responses between young and middle-aged adults. Considering the middle-aged group alone, the current protocol elicited similar trunk kinematics to what other protocols prompted. In studies with older adults, the peak trunk flexion angles from recovery trials were 22.0˚ to 37.3˚ [13, 42]; in our study, the averaged peak trunk flexion angle in the middle-aged to older group are 19.9˚ and 33.7˚ under small and large perturbations respectively. Comparing to young adults, more falls are induced in the middle-aged group. Additionally, the middle-aged group demonstrates greater peak trunk flexion angle. Gait speed has been identified as an indicator of balance, mobility, and function in older adults, but an individual may not walk significantly slower until entering their 70s [43, 44]. The middle-aged group in our study consists of middle-aged to older, asymptomatic adults (mean age = 57 years) who show similar comfortable walking speeds relative to their young counterparts (Table 1). Nevertheless, they perform significantly different when undergo our protocol. The reactive responses evoked by this protocol are more advanced than level walking. Utilizing this protocol could detect motor differences not shown in gait speeds in the two age groups.

In conclusion, the proposed protocol is successful in delivering perturbations at precise timings and invoked reliable responses. The two acceleration profiles simulate two levels of magnitude of tripping perturbations as intended. The protocol is capable of differentiating the trip recovery responses between young and middle-aged to older adults, suggesting its clinical utility.

## Supporting information

**S1 File. Step-by-step protocol, also available on protocols.io.** dx.doi.org/10.17504/protocols.io.ewov1o8k7lr2/v1.
(PDF)

## Acknowledgments

The authors would like to thank Szu-Ting Yi for the technical support, and thank Samuel Hadley, Catrina Fabian, James Anderson, and Denise Ng for their contribution to this project.

Associated content

dx.doi.org/10.17504/protocols.io.ewov1o8k7lr2/v1.

## Author Contributions

**Conceptualization:** Hui-Ting Shih, Szu-Ping Lee.

**Data curation:** Hui-Ting Shih.

**Formal analysis:** Hui-Ting Shih, Szu-Ping Lee.

**Funding acquisition:** Hui-Ting Shih, Robert Gregor, Szu-Ping Lee.

**Investigation:** Hui-Ting Shih, Szu-Ping Lee.

**Methodology:** Hui-Ting Shih, Szu-Ping Lee.

**Project administration:** Hui-Ting Shih, Szu-Ping Lee.

**Resources:** Hui-Ting Shih, Szu-Ping Lee.

**Software:** Hui-Ting Shih, Szu-Ping Lee.

**Supervision:** Robert Gregor, Szu-Ping Lee.

**Validation:** Hui-Ting Shih, Szu-Ping Lee.

**Visualization:** Hui-Ting Shih.

**Writing – original draft:** Hui-Ting Shih, Szu-Ping Lee.

**Writing – review & editing:** Hui-Ting Shih, Robert Gregor, Szu-Ping Lee.

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
