## [Decision Letter · Decision Letter 0]

21 Feb 2023

PONE-D-22-29207

Description, Reliability and Validity of a Ground-Reaction-Force Triggered Protocol for Precise Delivery of Unilateral Trip-Like Perturbations During Gait

PLOS ONE

Dear Dr. Lee,

Thank you for submitting your manuscript to PLOS ONE. After careful consideration, we feel that it has merit but does not fully meet PLOS ONE’s publication criteria as it currently stands. Therefore, we invite you to submit a revised version of the manuscript that addresses the points raised during the review process.

ACADEMIC EDITOR: 

The Reviewers suggested some modification to improve the quality of the paper. Please provide a detailed explanation of each point.  

A rebuttal letter that responds to each point raised by the academic editor and reviewer(s). You should upload this letter as a separate file labeled 'Response to Reviewers'.A marked-up copy of your manuscript that highlights changes made to the original version. You should upload this as a separate file labeled 'Revised Manuscript with Track Changes'.An unmarked version of your revised paper without tracked changes. You should upload this as a separate file labeled 'Manuscript'

We look forward to receiving your revised manuscript.

Kind regards,

Luca Russo, Ph.D.

Academic Editor

PLOS ONE

Journal Requirements:

Reviewers' comments:

Reviewer's Responses to Questions

**Comments to the Author**

1. Does the manuscript report a protocol which is of utility to the research community and adds value to the published literature?

Reviewer #1: Yes

Reviewer #2: Yes

2. Has the protocol been described in sufficient detail?

To answer this question, please click the link to protocols.io in the Materials and Methods section of the manuscript (if a link has been provided) or consult the step-by-step protocol in the Supporting Information files.

The step-by-step protocol should contain sufficient detail for another researcher to be able to reproduce all experiments and analyses.

Reviewer #1: Yes

Reviewer #2: Partly

3. Does the protocol describe a validated method?

Reviewer #1: Yes

Reviewer #2: No

4. If the manuscript contains new data, have the authors made this data fully available?

Reviewer #1: N/A

Reviewer #2: No

**5. Is the article presented in an intelligible fashion and written in standard English?**

Reviewer #1: Yes

Reviewer #2: Yes

6. Review Comments to the Author

Reviewer #1: Thank you for inviting me to review the article “Description, Reliability and Validity of a Ground-Reaction-Force Triggered Protocol for Precise Delivery of Unilateral Trip-Like Perturbations During Gait”.

With this Protocol Lab, the authors broaden the current knowledge on 'treadmill-based perturbation protocols', providing useful information to the research community.

The materials and methods are clear and well detailed, and the results are consistent with the purpose of the study.

I have some suggestions for improving the manuscript.

• There is no justification for the sample size.

• I suggest reviewing the units of measurement and reporting them as “m·s−1” instead of “m/s”

• Where possible, highlight the significant differences also in the tables.

• I suggest to the authors to broaden the 'Data Discussion' by also referring to what is already present in the literature.

Reviewer #2: I want to thank the Editors and the Authors for the chance to review this manuscript.

This study aims to develop and validate a treadmill-based protocol for generating unanticipated perturbations during walking.

The methodology is interesting and well-reported. However, the background and the discussion of the findings lack some details, potentially leaving the reader without compelling insights regarding the practical applicability of the study outcomes. I would suggest substantially improving and expanding these sections to increase the manuscript value.

In addition, as a practical note, I must report that the manuscript does not include line numbering, as suggested in the journal guidelines, making the review process more difficult.

The detailed comments are listed below:

Page 4: “Researchers have focused on trunk kinematics…”, please elaborate more on the past findings to provide a more comprehensive background.

Pages 4-5: what does “the precision of perturbation delivery” affect? The kinematics or the reaction response? Please describe in more detail the previous studies limitations that create a gap in the literature that the present study aims to fill.

Page 5: the purpose “to further improve biomechanical evaluations...” seems way too broad. Please, be more specific in formulating aims and hypotheses for the study.

Page 7: how was CWS assessed and set for each subject? Please explain.

Page 7: what does “more realistic magnitudes” mean in this context? It seems vague terminology.

Page 8: why the delay was quantified? Please explain

Page 8: I doubt whether studying kinematics differences between the two age groups may be considered a “validity” analysis. Demonstrating that there are differences would not mean that the protocol is valid per se. I would reword this particular purpose.

Page 9: How long the walking trials were? How many trials did each subject perform, and how were the kinematic data averaged and extracted?

Please provide more details.

Page 12: how was the trunk kinematics “consistency” between the visits assessed? Please explain.

Page 12: usually, older adults have a significantly lower preferred walking speed than younger. However, this study reports no differences. What do the Authors think about this result, and how can they explain it?

Page 13: the discussion is too short of discussing the interesting results effectively. Please elaborate more, providing a comparison to previously published results and practical applications.

The quality of the figures is low. They should be improved.

7. PLOS authors have the option to publish the peer review history of their article (what does this mean?). If published, this will include your full peer review and any attached files.

Reviewer #1: No

Reviewer #2: No

---

## [Author Response · Author response to Decision Letter 0]

8 Mar 2023

A point-by-point response to reviewers' comments is included in this revision submission. Please see the attached file "Response to reviewers v2.docx".

---

## [Decision Letter · Decision Letter 1]

29 Mar 2023

Description, Reliability and Utility of a Ground-Reaction-Force Triggered Protocol for Precise Delivery of Unilateral Trip-Like Perturbations During Gait

PONE-D-22-29207R1

Dear Dr. Lee,

We’re pleased to inform you that your manuscript has been judged scientifically suitable for publication and will be formally accepted for publication once it meets all outstanding technical requirements.

Kind regards,

Luca Russo, Ph.D.

Academic Editor

PLOS ONE

Additional Editor Comments (optional):

Congratulation for your achievement. Please read carefully the final comment of Reviewer 1 before to proceed with the final version of the manuscript.

Reviewers' comments:

Reviewer's Responses to Questions

**Comments to the Author**

1. Does the manuscript report a protocol which is of utility to the research community and adds value to the published literature?

Reviewer #1: Yes

Reviewer #2: Yes

2. Has the protocol been described in sufficient detail?

To answer this question, please click the link to protocols.io in the Materials and Methods section of the manuscript (if a link has been provided) or consult the step-by-step protocol in the Supporting Information files.

The step-by-step protocol should contain sufficient detail for another researcher to be able to reproduce all experiments and analyses.

Reviewer #1: Yes

Reviewer #2: Yes

3. Does the protocol describe a validated method?

Reviewer #1: Yes

Reviewer #2: No

4. If the manuscript contains new data, have the authors made this data fully available?

Reviewer #1: Yes

Reviewer #2: N/A

**5. Is the article presented in an intelligible fashion and written in standard English?**

Reviewer #1: Yes

Reviewer #2: Yes

6. Review Comments to the Author

Reviewer #1: The authors have responded satisfactorily to the reviewers' requests and the current version is improved and ready for publication.

The only note concerns some units of measurement (Lines 146-147 and 172) in which m s-2 was written instead of m s-1.

Reviewer #2: I want to thank the Authors for their thorough work addressing all my comments and recommendations. As a result, I believe the manuscript is significantly improved.

7. PLOS authors have the option to publish the peer review history of their article (what does this mean?). If published, this will include your full peer review and any attached files.

Reviewer #1: No

Reviewer #2: No

---

## [Editor Report · Acceptance letter]

13 Apr 2023

PONE-D-22-29207R1 

Description, Reliability and Utility of a Ground-Reaction-Force Triggered Protocol for Precise Delivery of Unilateral Trip-Like Perturbations During Gait 

Dear Dr. Lee:

I'm pleased to inform you that your manuscript has been deemed suitable for publication in PLOS ONE. Congratulations! Your manuscript is now with our production department. 

Kind regards, 

on behalf of

Dr. Luca Russo 

Academic Editor

PLOS ONE